# Overexpression of Reticulon 3 Enhances CNS Axon Regeneration and Functional Recovery after Traumatic Injury

**DOI:** 10.3390/cells10082015

**Published:** 2021-08-06

**Authors:** Sharif Alhajlah, Adam M Thompson, Zubair Ahmed

**Affiliations:** 1Neuroscience and Ophthalmology, Institute of Inflammation and Ageing, University of Birmingham, Birmingham B15 2TT, UK; alhjlah@su.edu.sa (S.A.); a.thompson@exeter.ac.uk (A.M.T.); 2Applied Medical Science College, Shaqra University, P.O. Box 1678, Ad-Dawadmi 11911, Saudi Arabia; 3Centre for Trauma Sciences Research, University of Birmingham, Edgbaston, Birmingham B15 2TT, UK

**Keywords:** Reticulon 3, spinal cord injury, optic nerve injury, axon regeneration, neurite outgrowth, protrudin

## Abstract

CNS neurons are generally incapable of regenerating their axons after injury due to several intrinsic and extrinsic factors, including the presence of axon growth inhibitory molecules. One such potent inhibitor of CNS axon regeneration is Reticulon (RTN) 4 or Nogo-A. Here, we focused on RTN3 as its contribution to CNS axon regeneration is currently unknown. We found that RTN3 expression correlated with an axon regenerative phenotype in dorsal root ganglion neurons (DRGN) after injury to the dorsal columns, a well-characterised model of spinal cord injury. Overexpression of RTN3 promoted disinhibited DRGN neurite outgrowth in vitro and dorsal column axon regeneration/sprouting and electrophysiological, sensory and locomotor functional recovery after injury in vivo. Knockdown of protrudin, however, ablated RTN3-enhanced neurite outgrowth/axon regeneration in vitro and in vivo. Moreover, overexpression of RTN3 in a second model of CNS injury, the optic nerve crush injury model, enhanced retinal ganglion cell (RGC) survival, disinhibited neurite outgrowth in vitro and survival and axon regeneration in vivo, an effect that was also dependent on protrudin. These results demonstrate that RTN3 enhances neurite outgrowth/axon regeneration in a protrudin-dependent manner after both spinal cord and optic nerve injury.

## 1. Introduction

Reticulons (RTNs) are membrane-bound proteins of which four have been identified in mammals and termed RTN1, 2, 3 and 4 (Nogo). RTNs are characterised by highly conserved reticulon homology domains (RHD); approximately 150–200 amino acids are located in the RTN C-terminal, which are involved in the improvement of the localisation and function of these proteins. RTNs are found in a variety of tissues including the brain, kidney, spleen and liver, playing diverse roles such as ER shaping and morphology, membrane trafficking and apoptosis. Moreover, RTNs are found in neurodegenerative diseases such as Alzheimer’s, amyotrophic lateral sclerosis and multiple sclerosis [1]. RTN1, 2 and 4 have three splice isoforms A, B and C whilst RTN3 only has two spliced isoforms RTN3A and B [2]. RTN3 is expressed in many types of tissues but was first isolated from the human retina and later shown to have the highest expression in the brain [3]. The human RTN3 gene is localised in chromosome 11q13 and mutations of this gene cause dysfunction of the retina and is, therefore, speculated that this gene may play a role in retinal function [4]. Furthermore, RTN3 is also localised in cultured cortical neurons and concentrated in axon growth cones in the early days after culture [4]. 

In neurons, a specific transport system for membrane-bound vesicles that contain lipids and proteins is required for the extension and maintenance of neurites [5,6]. Kinesin-1 is a molecular protein that plays a key role in this process, moving unidirectionally over long distances to mediate anterograde transport of neuronal cargos from the cell body to the synapse [7,8,9,10]. Kinesin-1 contains a dimer of kinesin heavy chain isoform 5A (KIF5) which hydrolyses adenosine triphosphate (ATP) and uses the resulting energy to move along microtubules [11]. Protrudin was identified as a key regulator of Rab11-dependent vesicular transport during neurite extension through the interaction of the atypical Rab11-binding domain with guanosine diphosphate (GDP)-bound form of Rab11 [12]. Protrudin is associated with Rab11 in PC12 cells in response to protrudin phosphorylation, resulting from signalling triggered by an extracellular signal-related kinase (ERK) in response to nerve growth factor [12]. In addition, expression of protrudin, even in non-neuronal cells, induced the formation of membrane protrusions whilst RNA interference (RNAi)-mediated downregulation of protrudin, inhibited neurite formation [12]. Interestingly, protrudin associates with KIF5 and its overexpression in HeLa cells induced the formation of protrusions similar to those observed after protrudin overexpression [13].

Protrudin also interacts with RTN3 and promotes its binding to KIF5 allowing the protrudin-KIF5 complex to cargo RTN3 from the cell soma to neurites [13]. More recently, protrudin was shown to form repeated contact sites with late endosomes (LEs; the main organelles of the secretory and endocytic pathways along with the endoplasmic reticulum (ER)) via coincident detection of the small GTPase RAB7 and phosphatidylinositol 3-phosphate (PtdIns(3)P), mediating transfer of the microtubule kinesin 1 from protrudin to the motor adaptor FYCO1 (FYVE and coiled-coil domain autophagy adaptor 1) on LEs [14]. These repeated LE-ER contacts promote microtubule-dependent translocation of LEs to the cell periphery and subsequent synaptotagmin-VII (SYT7)-dependent fusion with the plasma membrane, promoting neurite outgrowth [14]. In this study, we determined the contribution of RTN3 to CNS axon regeneration in a well-characterised dorsal column (DC) model of spinal cord injury (SCI) [15,16,17,18,19]. We then employed a second model of CNS injury, optic nerve crush (ONC) injury [19,20] to confirm our findings in the SCI model.

## 2. Materials and Methods

### 2.1. Animals

All experiments were licensed by the UK Home Office (PDA74B4CC6) and experimental protocols were approved by the University of Birmingham’s Animal Welfare and Ethical Review Board (AWERB; date of approval 27 October 2015). All animal surgeries were carried out in strict accordance with the guidelines of the UK Animals Scientific Procedures Act, 1986 and the Revised European Directive 1010/63/EU. The ARRIVE guidelines were also followed in the reporting of in vivo studies. Surgical procedures conformed to the guidelines and recommendation of the use of animals by the Federation of the European Laboratory Animal Science Associations (FELASA). Adult 6–8-week-old female Sprague-Dawley rats weighing 170–220 g (Charles River, Margate, UK) were housed in dedicated facilities at the University of Birmingham. Up to four rats per cage were maintained in a temperature-controlled environment on a 12 h light/dark cycle with free access to food and water ad libitum. All in vitro and in vivo experiments were performed with the experimenter blinded to the treatment conditions and unblinded after data analysis.

### 2.2. In Vitro Studies

#### 2.2.1. Preparation of CNS myelin Extracts (CME)

CME was prepared from adult rat brains as described by us previously [21]. Briefly, brains were homogenised in sucrose/EDTA (Sigma, Poole, UK) and debris was clarified by centrifugation. The supernatant was layered on 0.9 M sucrose and centrifuged at 20,000 *g* for 60 min. The white material at the interface was collected and purified in 0.32 M sucrose solution and the white pellet was collected and freeze-dried overnight. The protein content of the CME was determined using the Pierce BCA assay (#5000001; BioRad, Watford, UK) and the content of myelin-derived axon growth inhibitory molecules such as myelin basic protein (#MCA409S; MBP; Serotec, Oxford, UK; 1:500 dilution), Nogo-A (#AB588; Sigma; 1:500 dilution), myelin-associated glycoprotein (#MAB1567; MAG; Sigma; 1:500) and the CSPG brevican (#SC-135849; Santa Cruz Biotechnology, Santa Cruz, CA, USA; 1:500 dilution) was determined by Western blot, as described below.

#### 2.2.2. Adult Rat Primary DRGN Cultures

Rat primary DRGN were prepared according to our previously published method [21]. Briefly, rats were killed by CO_2_ overexposure before harvesting L4/L5 DRG pairs. Cells were dissociated into single cells using collagenase (#C9891; Sigma) and DRGN were grown in Neurobasal-A (NBA; #10888-022, Invitrogen, Paisley, UK) at a plating density of 350 cells/well in 8-well chamber slides precoated with 100 μg/mL poly-D-lysine (#A-003-M, Sigma). CNS myelin extracts (CME) were prepared and added to cultures at 100µg/mL to simulate the post-injury environment of the injured spinal cord and incubated with or without appropriate treatments for 3 days at 37 °C and 5% CO_2_, as described previously [21]. Experiments were performed in triplicates and repeated on 3 independent occasions (*n* = 9 wells/condition). 

#### 2.2.3. Primary DRGN Cultures after Preconditioning SN Lesion 

Since preconditioning SN (pSN) lesioned DRGN contain significantly higher levels of RTN3 mRNA/protein, we exposed the left SN at mid-thigh level and crushed the nerve using fine forceps at the level of the sacrotuberous ligament. We then harvested the ipsilateral L4/L5 DRGs at 7 days after injury and dissociated them into single cells and cultured them as described above. 

#### 2.2.4. Adult Rat Primary Retinal Cell Cultures

Retinal cells were dissociated into single cells using a Papain dissociation kit (#LK003150, Worthington Biochemicals, Lakewood, NJ, USA), as described by us previously [20]. Briefly, 125 × 10^3^ cells were plated in 8-well chamber slides in NBA pre-coated with 100 μg/mL poly-D-lysine (#P6407; Sigma) and CME and incubated with or without appropriate treatments for 3 days at 37 °C and 5% CO_2_ [20]. Experiments were performed in triplicates and repeated on 3 independent occasions (*n* = 9 wells/condition). 

#### 2.2.5. DRGN/RGC Neurite Outgrowth and Survival

DRGN or retinal cells were fixed in 4% paraformaldehyde (PFA) (#F003; TAAB Laboratories, Aldermaston, UK), washed in several changes of PBS and immunocytochemistry performed as described previously [20,21]. Briefly, DRGN/RGC were stained with βIII-tubulin antibodies (#T8578 Sigma; diluted 1:200) to label the DRGN/RGC somata and neurites. With an investigator masked to the treatment conditions, each well of a chamber slide was split into 9 quadrants and images were captured randomly using an Axioplan 200 epi-fluorescent microscope, equipped with an Axiocam HRc and Axiovision Software (all from Carl Zeiss, Hertfordshire, UK). Axiovision was then used to measure the length of the longest DRGN neurite from at least 30 DRGN/well and all RGC/well. The number of DRGN/RGC with neurites and the total number of surviving DRGN were also recorded as described previously [20,21]. 

#### 2.2.6. Overexpression of RTN3 in DRGN and Retinal Cultures

To overexpress RTN3 in DRGN and retinal cells in vitro, a CMV promoter-driven rat RTN3 plasmid was purchased from Origene, Rockville, MD, USA (#RR200211L3). Transfection controls were carried out using a CMV driven, non-targeting, EGFP plasmid (#PS100065; Origene). In preliminary experiments, optimal concentrations of RTN3 plasmids were determined as 2 µg. In vivo-jetPEI (#201; referred to as PEI; Polyplus Transfection, New York, NY, USA) was prepared according to the manufacturer’s instructions and used to deliver 2 µg of RTN3 (PEI-RTN3) or EGFP (PEI-EGFP) plasmids were used to transfects DRGN and retinal cells. Additional controls included untreated DRGN/retinal cells or PEI alone. DRGN and retinal cells were allowed to incubate for 3 days prior to harvesting of cells and extraction of total protein/RNA for validation of RTN3 protein/mRNA overexpression.

#### 2.2.7. Knockdown of RTN3, Protrudin, FYCO1, RAB7 and SYT7 Using siRNA 

For siRNA transfection, a preliminary dose determination experiment was run and 5 nM siRTN3 delivered using Lipofectamine ((#11668027; LF2000; Invitrogen) was used to determine optimal knockdown of RTN3 mRNA/protein in DRGN/RGC cultures, as described previously [22]. The efficiency of knockdown was assessed using quantitative RT-PCR (qRT-PCR) or Western blot, as described below. RTN3 was knocked down in DRGN/retinal cultures using a siRNA to rat RTN3 (siRTN3; Assay ID 251374, cat no. AM16708, ThermoFisher Scientific, Leicester, UK). Protrudin (siProtrudin; Assay ID 196855, cat no. AM16708), FYCO1 (siFYCO1; Assay ID, s236101, cat no. 4390771, ThermoFisher Scientific), Rab7 (siRAB7; Assay ID 52712, cat no. AM16708, ThermoFisher Scientific), SYT7 (siSYT7; Assay ID 199167, cat no. AM16708, ThermoFisher Scientific) was used to knock down appropriate targets in DRGN/retinal cultures, delivered using LF2000. An siRNA to EGFP was also used as a non-targeted control for transfections [22]. Experiments were performed in triplicate and repeated on 3 independent occasions (*n* = 9 wells/condition). 

#### 2.2.8. Quantitative RT-PCR (qRT-PCR) in DRGN and Retinal Cultures

DRGN and retinal cells were washed with PBS and total RNA (*n* = 6–9 wells/treatment) was extracted using Trizol reagent according to the manufacturer’s instructions (Invitrogen). Appropriate primers were used to validate each gene of interest using the specific primers (Appendix A; [16]) or pre-validated primer sequences (Appendix A). qRT-PCR was performed on a LighCycler Q-PCR machine (Roche, Burgess Hill, UK) following published methods [23]. Fold changes were computed using the ΔΔCt method.

#### 2.2.9. Tissue/Cell Lysis, Western Blot and Densitometry

Total protein was extracted from L4/L5 DRG, DRGN cultures, retinal cells and retinae after lysis in ice-cold lysis buffer as appropriate and subjected to Western blot as described previously [20,21]. A DC protein assay kit (#5000111, Bio-Rad, Watford, UK) was then used to determine total protein content and 40 µg of total protein from each relevant sample was separated on 12% SDS-PAGE, transferred onto PVDF membranes (#88520; ThermoFisher Scientific) and probed for RTN3 (rabbit polyclonal anti-RTN3; #ABN1723; Merck Millipore, Watford, UK; diluted 1:500); Protrudin (Rabbit anti-Protrudin; #MBS8570624; MyBiosource, San Diego, CA, USA; diluted 1:500); FYCO1 (Rabbit anti- FYCO1; #MBS9600315; MyBiosource; diluted 1:500); RAB7 (Rabbit anti-RAB7; #ab137029; Abcam, Cambridge, UK; diluted 1:500); and SYT7 (Mouse anti-SYT7; #MBS800258; MYBioscource) in 5% non-fat milk. Bands were detected using HRP-labelled anti-rabbit IgG and developed using an enhanced luminescence kit (#GENA934; GE Healthcare, Watford, UK). Beta-actin (#A5316; Sigma; 1:1000) was used as a loading control. An investigator masked to the treatment conditions quantified bands from 3 repeats by densitometry using the built-in-gel plotting macros in Image J (https://imagej.nih.gov/ij/index.html, accessed on 24 October 2020), as described previously [24]. 

### 2.3. In Vivo Studies

#### 2.3.1. Experimental Design

Animals were randomly assigned to treatment groups and experimenters were masked to the treatment conditions. Sample sizes were not computed using power calculations, instead, these were based on previous similar experiments published by us [16,19]. Unless otherwise stated, all in vivo experiments were performed with *n* = 4 rats/group/test and each experiment repeated on at least 2–3 independent occasions (total *n* = 8–12 rats/group), except for the electrophysiology, ladder crossing and tape sensing/removal tests where *n* = 6 rats/group were used and repeated on 3 independent occasions (total *n* = 18 rats/group). The same rats were used for electrophysiology, ladder crossing and tape sensing/removal tests. 

For the in vivo experiments, 5 groups of animals each comprised of 4 adult female Sprague-Dawley rats were designated as: (1), uninjured intact controls (IC); (2) sham-treated controls (Sham: partial laminectomy but no DC crush injury) (3), non-regenerating DC model (DC) that received a DC lesion alone; (4), regenerating SN model (SN) that received an SN lesion alone; and (5), regenerating pre-conditioning (p) SN lesion (pSN) + DC injury model (pSN + DC-pSN lesion performed 7 days before the DC lesion). The ipsilateral L4/L5 DRG were harvested from each animal at 10 days after DC, SN and pSN + DC lesions, pooled together and RNA and protein extracted for qRT-PCR and Western blot, respectively (*n* = 16 DRG/group (4 rats/group, 2 independent repeats)) and immunohistochemistry studies (*n* = 16 DRG/group (4 rats/group, 2 independent repeats)). 

To stimulate the intrinsic capacity of DRGN to grow neurites in vivo (‘regeneration readiness’), rats received a unilateral pSN lesion 7 days prior to harvesting of affected L4/L5 DRG [25,26]. In vivo stimulated DRGN were then dissociated into single cells using collagenase, cultured and treated as described in the appropriate section below. 

#### 2.3.2. RTN3 Overexpression

To overexpress RTN3 in DRGN or retina in vivo, a CMV promoter-driven rat RTN3 plasmid was purchased from Origene, Rockville, MD, USA (Cat no. RR200211L3). A CMV driven, non-targeting, EGFP plasmid (Cat No. PS100065) was used as control. Plasmid DNA for either RTN3 or the control EGFP vector were delivered by intra-DRG injection into L4/5 DRG or intravitreal injection into the eye, respectively, condensed in in vivo JetPEI (PEI; #201, Polyplus Transfection), a non-viral vector that we have shown to be as efficient at transducing neurons as efficiently as AAV8 [16,27]. PEI was prepared according to the manufacturer’s instructions and in preliminary experiments, 2 µg of plasmid DNA was determined to be optimal for RTN3 overexpression in vitro and in vivo (not shown). An equivalent amount of the GFP plasmid was used to control for RTN3 transfection. In further experiments pre-optimised concentrations of plasmids were then used to monitor the time-frame of RTN3 overexpression in DRGN and retina in vivo at 72 h, 1 week, 2 weeks, 4 weeks and 6 weeks by qRT-PCR, and at 14 days by Western blot and immunohistochemistry after intra-DRG or intravitreal injection. PEI maintains overexpression for at least 6 weeks in both DRGN and retina. 

To test the potential of RTN3 to promote DC axon regeneration and improve functional recovery, groups of 6 adult Sprague-Dawley rats (total *n* = 18 rats/group from 3 independent repeats) received pSN lesions 7 day before DC lesion to stimulate regeneration ’readiness’. Immediately after DC injury at day 0, animals received an intra-DRG injection of in vivo-jetPEI (referred to as PEI from herein), PEI-GFP, or PEI-RTN3 and were allowed to recover for 6 weeks during which behavioural tests were performed at baseline, 2 days and then weekly at 1–6 weeks after injury [16,28]. Electrophysiology to record compound action potentials (CAP) was also performed on the same set of animals at 6 weeks after injury before harvesting tissues for immunohistochemistry and determination of axon regeneration by GAP43 (#33-5000; mouse monoclonal anti-GAP43 antibody; Invitrogen; 1:400 dilution) immunohistochemistry [16]. 

Some animals (*n* = 6/group) also received 0.5 µL of the bi-directional tracer, FluoroRuby ((5% in saline) #D1820; MW 10,000; ThermoFisher Scientific), injected slowly using a micropipette with a tip diameter between 30–40 µm, at a depth of 0.5 mm and 10 mm caudal to the lesion site 1 day before animal sacrifice. Rats were then intracardially perfused with 4% paraformaldehyde, cryoprotected in sucrose, embedded in optimal cutting temperature (OCT; Raymond A Lamb, Peterborough, UK) embedding medium and longitudinal sections of the spinal cord cut at a thickness of 15 µm using a cryostat, as described in the immunohistochemistry section below. 

For RGC axon regeneration, groups of 4 adult Sprague-Dawley rats (3 independent repeats; total *n* = 12 rats/group) received intravitreal injection of RTN3 or the control EGFP vector and were allowed to recover for 28 days after ONC. Optic nerve regeneration was then assessed using the gold standard method in the rat of GAP43 immunohistochemistry, as described later. 

#### 2.3.3. Protrudin shRNA In Vivo

DNA fragments encoding stem-loop–type shRNAs specific for human protrudin mRNA (5′-GCTGAGGTGAAGAGCTTCTTG-3′) or GFP (non-specific control) mRNA (5′ GCTGACCCTGAAGTTCATC-3′) were synthesised, attached to the U6 promoter and subcloned into pBluescriptII SK(+) vector (#212205; Stratagene, La Jolla, San Diego, CA, USA), as described previously [13]. The efficiency of knockdown was assessed by the introduction of various concentrations of the plasmid vectors using in vivo-JetPEI by intra-DRG (L4/5) or intravitreal injection in vivo, as described above. In preliminary experiments, the optimal concentration of plasmid DNA to knock down Protrudin in vivo was determined as 2 µg (not shown). Knockdown efficiency with preoptimised concentrations of plasmids was assessed at 72 h, 1 week, 2 weeks, 4 weeks and 6 weeks (*n* = 4 DRG/time-point (2 rats/time-point), 3 independent repeats (total *n* = 12 DRG/time-point (6 rats/time-point) or 6 retinae/timepoint), by qRT-PCR analysis, with cells expressing control non-targeting EGFP shRNA. 

#### 2.3.4. Dorsal Column (DC) Crush Injury

For DC and SN crush lesions, rats received subcutaneous Buprenorphine injections before anaesthesia (5% isoflurane with 1.8 L/min O_2_). After a partial laminectomy, DC were crushed bilaterally at the level of T8 using calibrated watchmaker’s forceps [29,30,31]. The tips of the forceps, separated to a width of 1 mm, were inserted into the cord through the dorsal meninges to a depth of 1 mm and the DC crushed by tip approximation. For SN or pSN lesion, the left SN was exposed at mid-thigh level and crushed using fine forceps at the level of the sacrotuberous ligament. In the pSN + DC model, SN were crushed 1 week before DC crush injury. Pre- and post-surgical analgesia was used as standard and as advised by the Named Veterinary Surgeon.

#### 2.3.5. FluoroRuby Tracer Injection

Since Cholera toxin B labelling does not work in the rat in our hands (Ahmed et al., 2014), we used FluoroRuby (FR) bidirectional tracer to confirm regenerating axons in the DC. A micropipette with a tip diameter between 30–40 µm was used to inject 0.5 µL of 5% FR (#D1820; MW 10,000; ThermoFisher Scientific) dissolved in 0.9% saline at a depth of 0.5 mm, 10 mm caudal to the lesion site 1 day before animal sacrifice. Rats were then intracardially perfused with 4% paraformaldehyde, cryoprotected in a graded series of sucrose, embedded in optimal cutting temperature (OCT; Raymond A Lamb) embedding medium and longitudinal sections of the spinal cord cut at a thickness of 15 µm using a cryostat, as described in the immunohistochemistry section below. The same sections were stained for GAP43 and FR to demonstrate co-staining, as described below. 

#### 2.3.6. Optic Nerve Crush (ONC) Injury

After injection of rats with Buprenorphine to provide analgesia, rats were anaesthetised using 5% isoflurane in 1.8 L/min of O_2_ with constant body temperature and heart rate monitoring. Optic nerves were unilaterally crushed 2 mm from the lamina cribrosa using a calibrated watchmaker’s forceps, as described by us previously [21]. In general *n* = 4 rats/group were used and each experiment was repeated on 3 independent occasions (total *n* = 12 rats/group).

#### 2.3.7. FluroGold Back-Labelling of RGC

FluorGold (FG) retrograde tracer was used to count the number of surviving RGC in retinal wholemounts as described by us previously [32]. Briefly, 2 days before killing rats, 4% FG (#BT80014; Cambridge Bioscience, Cambridge, UK) retrograde tracer was injected into the nerve mid-way between the lamina cribrosa and the site of ONC. After 48 h, animals were killed by CO_2_ overexposure and animals were intracardially perfused with 4% formaldehyde, retinae dissected out, immersion fixed and wholemounted onto glass slides. Images were captured from three different areas of each retinal quadrant (total *n* = 12 counts per quadrant, *n* = 4 retinas per treatment, to account for the variation of RGC numbers in the different areas and quantified using the built-in particle counting facilities in ImagePro; Media Cybernetics, Bethesda, Rockville, MD, USA) and expressed as the number of RGC per mm^2^ ± S.E.M.

#### 2.3.8. Tissue Harvesting and Processing

For the microarray and Western blot studies, rats were killed at 10 days after DC lesion by CO_2_ overexposure and freshly isolated L4/L5 DRG, spinal cord, retinae or optic nerves from appropriate experiments were harvested and snap-frozen in liquid N_2_ before RNA extraction for quantitative RT-PCR (qRT-PCR) or protein extraction for Western blot, as described later.

For immunohistochemistry, animals were intracardially perfused with 4% PFA (TAAB Laboratories) in phosphate-buffered saline (PBS; #P4417, Sigma, Poole, UK) and both ipsilateral and contralateral L4/L5 DRG and lesion site plus 5 mm either side of the spinal cord and retinae/optic nerves were dissected out and post-fixed for 2 h at 4 °C. All tissues were then cryoprotected in a graded series of sucrose, blocked up in optimal cutting temperature embedding compound (#F320/B1, TAAB laboratories) and sectioned through a para-sagittal plane on a cryostat set at 15 µm-thick. Sections from the entire DRG, spinal cord, retinae or optic nerves were collected onto charged slides and stored at −20 °C until required, noting the middle sections of each sample. 

#### 2.3.9. Quantitative RT-PCR (qRT-PCR)

The levels of RTN3 mRNA were validated by qRT-PCR using a pre-validated primer sequence (rat RTN3, Rn01498010_m1; cat no. 4351372; ThermoFisher Scientific, Loughborough, UK). Levels of protrudin mRNA were analysed using pre-validated primer sequences (rat Protrudin, RN01766356_m1; cat no. 4331182; ThermoFisher Scientific). Q-PCR was performed on a LighCycler Q-PCR machine (Roche, Burgess Hill, UK) following published methods [23]. Fold changes were computed using the ΔΔCt method.

#### 2.3.10. Immunohistochemistry

The middle sections of each DRG, spinal cord, optic nerve or retinae were chosen for immunohistochemistry as described by us previously [16,31,32,33]. One-three sections per biological replicates were analysed and quantified for each group, and the number of biological replicates per group is indicated in the figure legends. Briefly, masked slides were allowed to thaw at room temperature before being washed in PBS. Non-specific antibody binding was then blocked with 3% bovine serum albumin diluted in PBS containing 0.1% Triton X-100 before incubation with relevant antibody overnight at 4 °C in a humidified chamber. DRG were co-stained with rabbit polyclonal anti-RTN3 (#ABN1723; Merck Millipore; 1:400 dilution) and monoclonal anti-NF200 (#N5389; Sigma, 1:400) primary antibodies. RGC in the retina were co-stained with RTN3 as above and monoclonal anti-βIII-tubulin (#T8578; Sigma; 1:200). GAP43 was used to detect regenerating axons in the cord/optic nerve and detected with mouse anti-GAP43 (#33-5000 (clone 7B10); Invitrogen; 1:400) primary antibody. GAP43 was used since Cholera toxin B labelling in our hands did not label regenerating DC axons in the rat [31]. After several washes in PBS, Alexa488 anti-rabbit (#A323731; Invitrogen)/Texas Red anti-mouse (#PA1-28626; Invitrogen) (RTN3 and NF200+/βIII-tubulin staining) and Alexa488 anti-mouse (#A32723; Invitrogen) (to detect GAP43^+^ staining) were diluted 1:400 in PBS and incubated for 1 hr at room temperature in a humidified chamber. Sections were then washed in several changes of PBS and coverslips mounted using Vectashield containing DAPI (#H-1900; Vector Laboratories, Peterborough, UK). Negative controls, including omission of primary antibody, were included in each run and were used to set the background threshold level for each antibody before image capture. Slides were viewed using an Axioplan 200 epi-fluorescent microscope equipped with an Axiocam HRc and run using Axiovision Software (all from Zeiss, Herefordshire, UK) 

#### 2.3.11. Other Antibodies for Immunocytochemistry/Immunohistochemistry

The following antibodies were used for immunoassays: primary antibodies-rabbit polyclonal anti-RTN3 (#ABN1723; Merck Millipore; 1:400 dilution), monoclonal anti-NF200 (#N5389, Sigma, 1:400), monoclonal anti-βIII-tubulin (#T8578; Sigma; 1:200), mouse anti-GAP43 (#33-5000 (clone 7B10); Invitrogen; 1:400); secondary antibodies-Alexa488 anti-rabbit (#A323731; Invitrogen; 1:400 dilution), Texas Red anti-mouse (#PA1-28626; Invitrogen; 1:400 dilution), Alexa488 anti-mouse (#A32723; Invitrogen; 1:400 dilution).

#### 2.3.12. Quantification of GAP43^+^ Axons

Regenerating GAP43^+^ axons in the spinal cord were quantified by collecting all serial parasagittal 50 µm-thick sections (70–80 sections/rat; *n* = 6 rats/group, 3 independent repeats; total *n* = 18 rats/treatment) and counting the number of GAP43^+^ fibres through a dorsoventral oriented line from 4 mm rostral to 4 mm caudal to the lesion site, as described previously [16,34]. Axon number was calculated as a percentage of the number of GAP43^+^ fibres seen 4 mm above the lesion site where the DC was intact. 

Regenerating GAP43^+^ axons in the optic nerve were quantified from 3 sections/optic nerve (*n* = 4 optic nerves/group, 3 independent repeats; total *n* = 12 optic nerves/group) at 0.2, 0.5, 1, 1.5, 2, 2.5, 3 and 4 mm from the lesion epicenter. GAP43 in the optic nerve correlates with the number of axons detected by the anterograde tracer, Rhodamine B and thus accurately demonstrates regenerating RGC axons [33]. The cross-sectional width of the optic nerve was also recorded where axon counts were performed and used to calculate the number of axons/mm optic nerve width. The total number of axons extending distance d in an optic nerve with a radius r, Σad, was calculated using the formula:
Σad = πr2 × (average number of axons/mm width)/(section thickness, t, 0.015 mm)

#### 2.3.13. Electroretinography

Electroretinography (ERG) was performed as described by us previously [35]. Briefly, rats were dark-adapted for 12 h and after anaesthesia, eyes were dilated with tropicamide. Scotopic flash ERG was recorded under dim red light from −5.5 log cd·s/m^2^ to 1.0 log cd·s/m^2^ in 0.5 log unit increments. Traces were analysed using the built-in Espion software and traces at a light intensity of −5.5 log cd·s/m^2^ were chosen for analysis as they produced clean, unambiguous positive scotopic thresholds (pSTR) at 100 ms after stimulus. The peak amplitude of the pSTR was recorded by an experimenter masked to the treatment conditions.

#### 2.3.14. Electrophysiology

At 6 weeks after surgery or treatment, compound action potentials (CAP) were recorded after vehicle, RTN3 overexpression or RTN3 overexpression/Protrudin suppression (*n* = 6 rats/group, 3 independent repeats (total *n* = 18/group/test)) as previously described [16,36,37]. Briefly, with the experimenter masked to the treatment conditions, silver wire electrodes were used to apply single-current pulses (0.05 ms) through a stimulus isolation unit in increments (0.2, 0.3, 0.6, 0.8 and 1.2 mA) at L1/L2 and compound action potentials (CAP) recorded at C4/C5 along the surface of the midline spinal cord. CAP amplitudes were calculated between the negative deflection after the stimulus artefact and the next peak of the wave. CAP area was calculated by rectifying the negative component (full-wave rectification in Spike 2 software) and measuring its area at the different stimulation intensities. At the end of the experiment, the dorsal half of the spinal cord was transected between stimulating and recording electrodes to confirm that a CAP could not be detected. Spike 2 software (Cambridge Electronic Design, Cambridge, UK) was used to analyse the data and representative Spike 2 software processed CAP traces data are shown.

#### 2.3.15. Functional Tests after DC Injury

Functional testing was carried out as described by us previously [16,19,28,38]. Briefly, animals (*n* = 6 rats/group, 3 independent repeats (total *n* = 18/group/test)) received training to master traversing the horizontal ladder functional testing. Baseline parameters for all functional tests were established 2–3 days before injury. Animals were then randomly assigned and treatment status masked from the investigators and tested at 2 days after DC lesion + treatment and then weekly for 6 weeks. Experiments were performed by 2 observers masked to the treatment status, with animals being tested in the same order and at the same time of day. Three individual trials were performed each time for each animal for each test.

Horizontal ladder crossing test: This tests the animals’ locomotor function and is performed on a 0.9-m-long horizontal ladder with a diameter of 15.5 cm and randomly adjusted rungs with variable gaps of 3.5–5.0 cm. The total number of steps taken to cross the ladder and the number of left rear paw slips were recorded and the mean error rate was then calculated by dividing the number of slips by the total number of steps taken.

Tape removal test (sensory function): The tape removal test determines touch perception from the left hind paw. Animals were held with both hind-paws extended and the time it took for the animal to detect and remove a 15 × 15 mm piece of tape (Kip Hochkrepp #803, Bocholt, Germany) was recorded and used to calculate the mean sensing time.

### 2.4. Statistical Analysis and Data Visualisation

Data are presented as mean ± SEM. Normally distributed data were interrogated by one-way analysis of variance (ANOVA), with Dunnett’s post hoc tests, set at *p* < 0.05 in SPSS (IBM, NJ, USA). All *p* and *n* values are specified within each figure legend. No statistical tests were used to predetermine sample sizes, but our sample sizes were similar to those reported in previous publications [15,16,17,18,19]. All images shown are representative images of data quantified in corresponding graphs.

For the horizontal ladder test, data were analysed by comparing lesion versus sham-treated rats, using binomial generalised linear mixed models (GLMM), with lesioned/sham (‘LESION’; set to true in lesioned animals post-surgery, false otherwise) and operated/unoperated (‘OPERATED’; set to false before surgery, true after surgery) as fixed factors, animals as a random factor and time as a continuous covariate [16,19,38]. Data were fitted with lme4 with the *glmer* function and *p* values calculated using parametric bootstrap with *pkbrtest* in R package (www.r-project.org, accessed on 21 June 2019), set to between 1000 and 20,000 simulations, as described previously [16,28]. For the tape removal test, linear mixed models (LMM) were compared with *lme4* and *glmer* functions in the R package. *p* values were then calculated in R using *pkbrtest*, with the Kenward-Roger method [16,28].

## 3. Results

### 3.1. Levels of RTN3 Correlate with an Axon Regenerative Phenotype

To examine changes in RTN3 mRNA levels, we harvested dorsal root ganglion (DRG) containing the DRG neurons (DRGN) at 10 days after (1) DC injury (non-regenerating model); (2) sciatic nerve (SN) injury (regenerating SN model); (3) preconditioning (p) SN + DC lesion (crush of SN at −7 days before a DC lesion; regenerating DC model) (Figure 1A). Preconditioning lesions upregulate the intrinsic capacity of DRGN neurons to regenerate [17,26]. We used qRT-PCR to demonstrate that RTN3 levels were upregulated by 4.2 ± 0.05- and 6.2 ± 0.04-fold in regenerating SN and pSN + DC DRGN compared to those harvested after DC injury (Figure 1B). Western blot and subsequent densitometry (Figure 1C,D) confirmed that regenerating SN and pSN + DC models also contained significantly elevated RTN3 protein levels. Low levels of RTN3 immunoreactivity were detected in the cytoplasm and cell membranes of DRGN after DC injury (Figure 1E). However, in regenerating SN and pSN + DC DRGN, RTN3 immunoreactivity (green) was widely observed in the cytoplasm and cell membranes of DRGN after DC injury (Figure 1E). These results demonstrated that elevated levels of RTN3 correlated with an axon regenerative phenotype and that RTN3 was mainly localised in the cytoplasm of DRGN.

### 3.2. RTN3 Is Required for Enhanced DRGN Neurite Outgrowth

To understand the contribution of RTN3 to DRGN neurite outgrowth, we overexpressed RTN3 in freshly isolated primary adult DRGN using a non-viral vector, in vivo-JetPEI (called PEI from herein), which we have shown produces similar efficiencies of gene transduction as adeno-associated vectors (AAV8) in DRGN in vitro and in vivo [16]. Dissociated adult rat DRGN were allowed to settle overnight, prior to treatment with plasmids containing control EGFP (non-specific control) and RTN3, left to incubate in the presence of inhibitory concentrations of CNS myelin extracts (CME), prepared and used at concentrations as described by us previously [21] (Figure 2A). Western blot and subsequent densitometry confirmed nearly 3-fold overexpression of RTN3 in treated cultures, compared to EGFP-treated control cultures (Figure 2B,C). Overexpression of RTN3 disinhibited DRGN neurite outgrowth such that DRGN were significantly longer in length (Figure 2D,E) and the % of DRGN with neurites was also increased (Figure 2F). However, overexpression of RTN3 did not affect DRGN survival (Figure 2G).

Since preconditioning lesions (pSN + DC) stimulate significantly higher levels of RTN3 mRNA and protein in DRGN, we dissociated DRGN from animals that had received a preconditioning lesion at −7 days followed by a DC crush at day 0 and harvested DRGN 10 days later (Figure 3A). We then used PEI to knock down RTN3 using an shRNA plasmid DNA and grew them in the presence of neurite growth inhibitory concentrations of CME, to determine DRGN survival and disinhibited neurite outgrowth (Figure 3A). In a preliminary experiment, shRTN3 was titrated to determine that 2 µg of shRTN3 plasmids optimally caused 80% knockdown of RTN3 mRNA in cultured DRGN (Appendix A). Knockdown of RTN3 (with the optimal amount of shRNA plasmid DNA) reduced RTN3 mRNA and protein by 80% and 88%, respectively (Figure 3B–D). In DRGN cultures, knockdown of RTN3 almost completely ablates DRGN neurite outgrowth without affecting survival (Figure 3E–H). These results demonstrate that RTN3 is required for disinhibited DRGN neurite outgrowth.

We then determined that overexpression of RTN3 using PEI in preconditioned DRGN further upregulated RTN3 levels above that induced by preconditioning lesions alone (Figure 4A–D). Overexpression of RTN3 in DRGN cultures from pSN + DC-treated animals significantly potentiated DRGN neurite outgrowth in terms of both the mean neurite length and % DRGN with neurites, over and above that observed in pSN + DC cultures treated with a control EGFP vector (Figure 4E–G).

RTN3 overexpression has previously been shown to result in neurite abnormalities and reduction in synaptic plasticity in transgenic mice by the aggregation of RTN3 in RTN3 immunoreactive dystrophic neurites (RINDs) [39,40]. RINDs were abundantly present in a mouse model of Alzheimer’s disease (AD) and hence RTN3 overexpression has been implicated in causing hippocampal age-dependent RINDs. In our previous experiments, we did not observe any abnormalities in DRGN neurite outgrowth. We tested whether overexpression of RTN3 and subsequent exposure to Aβ1-42 resulted in DRGN abnormalities. However, overexpression of RTN3 and exposure to Aβ1-42 had no significant effects on disinhibited DRGN neurite outgrowth, DRGN survival nor were there any abnormalities in DRGN neurite formation, such as dystrophic neurites (Appendix A). These results demonstrate that DRGN does not suffer adverse effects of RTN3 overexpression as reported for hippocampal neurons.

Taken together, these results demonstrate that RTN3 overexpression promotes DRGN neurite outgrowth in naïve DRGN cultures. DRGN exposed to preconditioning lesions grow significantly longer neurites than naïve cultures and overexpression of RTN3 in preconditioned DRGN further promotes disinhibited DRGN neurite outgrowth. Since knockdown of RTN3 suppressed DRGN neurite outgrowth, this suggests that DRGN require RTN3 for neurite outgrowth.

### 3.3. RTN3-Mediated DRGN Neurite Outgrowth Requires Protrudin Signaling

We next sought to understand how RTN3 may signal to enhance DRGN neurite outgrowth. Since the interaction of RTN3 with protrudin and protrudin to RAB7, FYCO1 and SYST7 is important in regulating membrane protrusions resembling neurites, we overexpressed RTN3 in preconditioned (pSN + DC injured) DRGN and treated cultures with appropriate siRNAs to determine the contribution of protrudin, FYCO1, RAB7 and SYST7 to disinhibited DRGN neurite outgrowth (Figure 5A). Optimal concentrations of siRNAs to protrudin, RAB7, FYCO1 and SYST7 nearly ablated all detectable levels of the relevant proteins in DRGN cultures (Figure 5B–E) and completely suppressed DRGN neurite outgrowth observed after RTN3 overexpression (Figure 5F–K), without affecting DRGN survival (Figure 5L). These results suggest that RTN3-mediated DRGN neurite outgrowth requires protrudin or its binding partner signalling.

As off-target effects of siRNA transfection such as activation of interferon responses have been shown by us and others and appear to be siRNA sequence-dependent, we checked for a panel of interferon-mediated cytokines and innate immunity-related related genes in our cultures after knockdown of protrudin, FYCO1, RAB7 and SYT7 by qRT-PCR. We did not observe any activation of TNFα, IFNγ, IL-6, IL-12, IL1β, MX1, IFIT, OAS1 or Casp7 mRNA in RTN3-overexpressed DRGN. In contrast, transfection of RTN3-ver expressed DRGN cultures with a positive control sip75NTR, which we know invokes a significant interferon response [23], induced 12–17-fold increases in mRNA of all of these genes (Appendix A).

### 3.4. Overexpression of RTN3 Promotes DC Axon Regeneration and Functional Recovery In Vivo

To determine if RTN3 overexpression promotes DC axon regeneration in vivo and whether knockdown of Protrudin with shRNA (shProtrudin) ablates RTN3 overexpression-mediated axon regeneration over 6 weeks (Appendix A), we stained appropriately treated spinal cords with GAP43 antibodies to mark regenerating axons, since cholera toxin B labelling does not work in our hands in the rat [31]. Many GAP43^+^ axons were seen crossing the lesion site (*) and reaching at least 6 mm (>20%) into the distal segment of the spinal cords of pSN + DC + PEI-RTN3-treated rats compared to little or no GAP43^+^ staining in pSN + DC + PEI-EGFP-treated control spinal cords (Figure 6A,B). FluoroRuby (FR) anterograde tracer was used as a second method to confirm that GAP43^+^ regenerating axons were also co-labelled with FR in the same sections. However, treatment with shProtrudin at the same time as RTN3 overexpression (pSN + DC + PEI-RTN3 + shProtrudin) ablated GAP43^+^ regenerating axons in spinal cords, normally observed after RTN3 overexpression (pSN + DC + PEI-RTN3 + shEGFP) (Figure 6A,B).

To determine if overexpression of RTN3 led to functional benefits, we used electrophysiology, tape sensing and removal (sensory function) and ladder crossing tests (locomotor function) [16,18,19]. Individual CAP traces, CAP amplitudes and CAP areas that are normally ablated after DC injury were significantly improved in RTN3 overexpressed animals compared to controls (Figure 7A). However, RTN3-induced improvements in CAP traces, amplitudes and areas, were ablated when Protrudin was knocked down (Figure 7A–C). Overexpression of RTN3 also promoted significant improvements in sensory and locomotor function but again, these improvements were ablated if Protrudin was also knocked down (Figure 7D,E).

These results demonstrate that overexpression of RTN3 promotes DC axon regeneration after injury with a return of significant beneficial sensory and locomotor function.

### 3.5. Overexpression of RTN3 Promotes Upregulation of Regeneration Associated Genes after DC Injury

We overexpressed RTN3 in preconditioned DRGN using PEI-delivered plasmids and waited 10 days before harvesting DRG bundles and subjecting them to GAP43 immunostaining and qRT-PCR for regeneration associated genes (RAGs). We observed that overexpression of RTN3 led to increased numbers of RTN3^+^ DRGN (Appendix A), whilst qRT-PCR demonstrated significantly increased expression of RAGs, namely *gap43*, *sprr1a*, *cebp*/*epsilon*, *atf3* and *galanin* (Appendix A). This suggests that RTN3 overexpression in DRGN leads to activation of RAGs, which may account for the observed enhanced axon regeneration and functional recovery seen after DC injury and RTN3 overexpression.

### 3.6. Overexpression of RTN3 Promotes RGC Survival and Neurite Outgrowth

We then employed a second CNS injury model to assess the contribution of RTN3 to disinhibited neurite outgrowth/axon regeneration. We used the visual system, an extension of the CNS where optic nerve crush (ONC) injury leads to rapid death of retinal ganglion cells (RGCs) and axon regeneration can be promoted and easily visualised [19,20,41,42,43]. First, we used dissociated adult rat retinal cells, enriched in RGCs, overexpressed RTN3 and cultured them for 3 days in the presence of inhibitory concentrations of CME [20]. We then performed a Western blot to confirm that RTN3 was overexpressed by >3-fold in cells treated with RTN3 plasmids than in EGFP-vector controls (Appendix A). Staining for βIII-tubulin to mark RGC soma and neurites demonstrated that RTN3-overexpression promoted significantly longer RGC neurites (Appendix A) as well as increasing the % of RGC with neurites (Appendix A), compared to controls (Neurobasal-A (NBA) + shEGFP) or the positive control, ciliary neurotrophic factor (CNTF)) [20]. Interestingly, quantification of the number of βIII-tubulin^+^ RGC demonstrated that overexpression of RTN3 was significantly more neuroprotective than the control shEGFP plasmids, and as neuroprotective as the positive control, CNTF (Appendix A). These results demonstrate that overexpression of RTN3 in retinal cultures promotes RGC neurite outgrowth and survival.

### 3.7. Overexpression of RTN3 Promotes RGC Survival and Axon Regeneration after Optic Nerve Crush (ONC) Injury

Finally, we assessed the contribution of RTN3 overexpression in RGC survival and axon regeneration after optic nerve injury in vivo. Firstly, intravitreal injection of PEI-RTN3 plasmids in vivo led to an almost 3-fold overexpression of RTN3 in the retina (Figure 8A,B). Immunohistochemical staining with RTN3 antibodies demonstrated localisation of RTN3, primarily to the ganglion cell layer (GCL) and inner plexiform layer in ONC + shEGFP-treated control eyes, spreading to the inner (INL) and outer nuclear layers (ONL) in ONC + RTN3-treated eyes (Figure 8C). High power images demonstrated RTN3 (red) localisation to βIII-tubulin^+^ RGC (green; arrows) in the GCL in both RTN3 and shEGFP vector control eyes (Figure 8D). RGC axon regeneration in the optic nerve was significantly enhanced by overexpression of RTN3 compared to shEGFP-vector control eyes, whilst co-expression of shProtrudin, attenuated RTN3-induced RGC axon regeneration (Figure 8E,F). Like the in vitro results above, overexpression of RTN3 also promoted significant RGC survival in vivo, 1294 ± 145 and 1287 ± 106 vs 447 ± 25 in ONC + RTN3 + shEGFP, ONC + RTN3 + shProtrudin and ONC + shEGFP, respectively (representing survival of 45% in ONC + RTN3 + shEGFP-treated eyes vs 20% in ONC + shEGFP-treated controls) with suppression of Protrudin having no effect of RTN3-induced RGC survival (Figure 8G,H). Furthermore, overexpression of RTN3 significantly improved photopic scotopic thresholds (pSTR), a measure of RGC function, with protrudin suppression attenuating RTN3-induced functional benefits (Figure 8I,J). These results demonstrate that overexpression of RTN3 promotes RGC axon regeneration, survival and improved function after ONC.

## 4. Discussion

In this study, we show that RTN3 levels correlated with an axon regenerative phenotype and that overexpression of RTN3 enhanced DRGN neurite outgrowth, which was dependent on protrudin function. We also show that overexpression of RTN3 in the DC model of SCI promotes significant DC axon regeneration, improved electrophysiological, sensory and locomotor function. Furthermore, we used a second model of CNS axon regeneration, the optic nerve crush (ONC) injury model, to show that RTN3 is also required for enhanced RGC neurite outgrowth in vitro and RGC axon regeneration in vivo, effects that were also dependent on protrudin. Moreover, overexpression of RTN3 was also RGC neuroprotective in vitro and after ONC in vivo.

It is increasingly clear that axon regeneration fails after CNS injury including SCI and ONC due to a multitude of factors, including the low intrinsic capacity of CNS neurons to regenerate their axons, the presence of a glial scar containing axon growth inhibitory molecules and the presence of axon growth inhibitory molecules in degenerating myelin [44,45,46,47]. Some members of the reticulon family have also been shown to inhibit CNS axon regeneration. For example, RTN4/Nogo-A can bind to NgR1 and p75NTR and LINGO1/AMIGO3 to form a tripartite receptor complex that transmits growth cone collapse signals into the cell via the Rho-ROCK pathway [48,49]. It was later shown that the extracellular/ER luminal portion of Nogo-A, the 66-loop termed Nogo-66 (NgR1) inhibited neurite outgrowth [48,50]. Genetic ablation or pharmacological inhibition of the Nogo-A-NgR1 interactions promote axon growth and behavioural recovery after SCI [3,51,52,53,54]. In addition, RTN1, 2 and 3 were reported not to inhibit axon regeneration [48].

However, we report here that upregulation of RTN3 correlated with both SN and pSN + DC axon regeneration and its overexpression enhanced DRGN neurite outgrowth/axon regeneration, leading to significant improvements in electrophysiological, sensory and locomotor function after DC injury in rats. We backed up these findings using a second CNS population of neurons, RGCs, which also normally fail to regenerate their axons due to the same problems as is apparent after SCI. However, significant RGC neurite outgrowth and axon regeneration were promoted by overexpression of RTN3 in vitro and in vivo, effects that were also protrudin dependent, suggesting a commonality between DRGN and RGC in their response to RTN3. In contrast to the DC injury model, RTN3 overexpression was neuroprotective for RGC both in vitro and after ONC in vivo, which was not the case for DRGN where no effects of DRGN survival were observed. This is probably explained by our observations that after ONC, significant RGC death occurs within 3 weeks after injury [21,22,32,55] whilst after DC injury, little or no DRGN death is observed [56]. Hence, RTN3 overexpression in DRGN has little or no protective effect as they rarely die in culture or after DC injury.

RTN3 overexpression and subsequent improvements in DRGN neurite outgrowth were dependent on protrudin and its interacting molecules since knockdown of either protrudin, RAB7, FYCO1 or SYT7 ablated RTN3-induced DRGN neurite outgrowth. Moreover, RTN3-induced improvements in DC axon regeneration and electrophysiological, sensory and locomotor function were also ablated in vivo by suppression of protrudin. These results suggest that RTN3 works through protrudin and its interacting molecules and that suppression of any of these ablates RTN3-induced axon regeneration and subsequent functional improvements. It is possible that protrudin facilitates increased translocation of LEs after RTN3 overexpression and hence mediates enhanced axon regeneration and sprouting, bringing about enhanced functional recovery. In addition, the sprouting of preserved fibres around the lesion site could also account for some of the observed benefits of RTN3 to functional recovery.

Whether this is a direct or indirect effect needs to be elucidated. However, protrudin normally forms contact sites with late endosomes (LEs) via RAB7, mediating the transfer of kinesin-1 from protrudin to the motor adaptor FYCO1 on LEs [5,6,7,8,9,10,11,12,13,14]. Repeated LE-ER contacts then promote microtubule-dependent translocation of LEs to the cell periphery and subsequent SYT-VII-dependent fusion with the plasma membrane, inducing neurite outgrowth [5,6,7,8,9,10,11,12,13,14]. Thus, the interaction of RTN3 with protrudin may be an additional requirement for kinesin-1-bound protrudin-containing LE to be loaded onto FYCO1 contact sites and translocation of LEs to the plasma membrane, promoting protrusion and neurite/axon outgrowth. In addition, RTN3 overexpression may enhance the handover of kinesin-1 that is bound to protrudin containing LEs and hence the movement of greater numbers of LEs to the plasma membrane, enhancing neurite outgrowth/axon regeneration. We, therefore, propose a working model of RTN3-induced neurite outgrowth/axon regeneration whereby overexpression of RTN3 allows the handover of kinesin-1 to occur more rapidly and hence greater numbers of LEs are available for forming protrusions and hence neurites/axons at the plasma membrane. As the LEs are attached to protrudin and the kinesin-1 motor proteins, ablation of protrudin ablates the handover of kinesin-1 hence neurite outgrowth/axon regeneration is ablated (Figure 9). However, more experiments are needed to determine the exact mechanisms of this proposed model.

The improvements in sensory and locomotor recovery after RTN3 overexpression in the spinal cord was remarkable since injured animals were indistinguishable from sham-treated animals by 4 weeks after DC injury, suggesting significant recovery. Even at 2 days after injury, there were significant improvements in sensory and locomotor function, which we are unable to explain at present since it is likely to be too soon for axonal regeneration/plasticity to account for these improvements. The improvements at 2 days after RTN3 overexpression may be due to enhanced survival of injured cells in the spinal cord or greater preservation of cord tissue architecture. This remains to be investigated. Moreover, we have not analysed any effects of RTN3 overexpression on astrocyte responses, microglia/macrophage activation status nor scarring and scar-related inhibitory molecules. It is possible that RTN3 causes changes in these parameters which then facilitates the enhanced axon regeneration and functional recovery. This is important to investigate in future experiments.

In addition, the promotion of axon regeneration and functional benefits in the DC injury, we have shown that overexpression of RTN3 also promotes RGC survival, neurite outgrowth, axon regeneration and restoration of significant RGC function by electroretinography in the ONC injury model. The ONC injury model is an excellent model to evaluate neuroprotective and axon regenerative therapies since RGC soma are easily accessible after intravitreal injection. Like our observations in the DC injury model, RGC axon regeneration promoted by RTN3-overexpression was dependent on protrudin function since knockdown of protrudin ablated RTN3-induced RGC neurite outgrowth/axon regeneration and functional benefits. Moreover, RTN3 had an RGC neuroprotective effect after ONC compared to DRGN after DC injury. This is probably because RGC death occurs rapidly after ONC whilst DRGN death rarely occurs after DC injury. Hence, RTN3 probably exerts a neuroprotective effect on both systems but is only evident after ONC due to the overt death of RGC that normally occurs.

## 5. Conclusions

This study supports the notion that overexpression of RTN3 enhances neuroprotection and CNS axon regeneration, and in particular DC and RGC axon regeneration and thus leading to improved sensory, locomotor and visual function. Our study, therefore, implies that overexpression of RTN3 may be a possible treatment for SCI and ONC injury and may restore lost function in affected patients.

## Figures and Tables

**Figure 1 cells-10-02015-f001:**
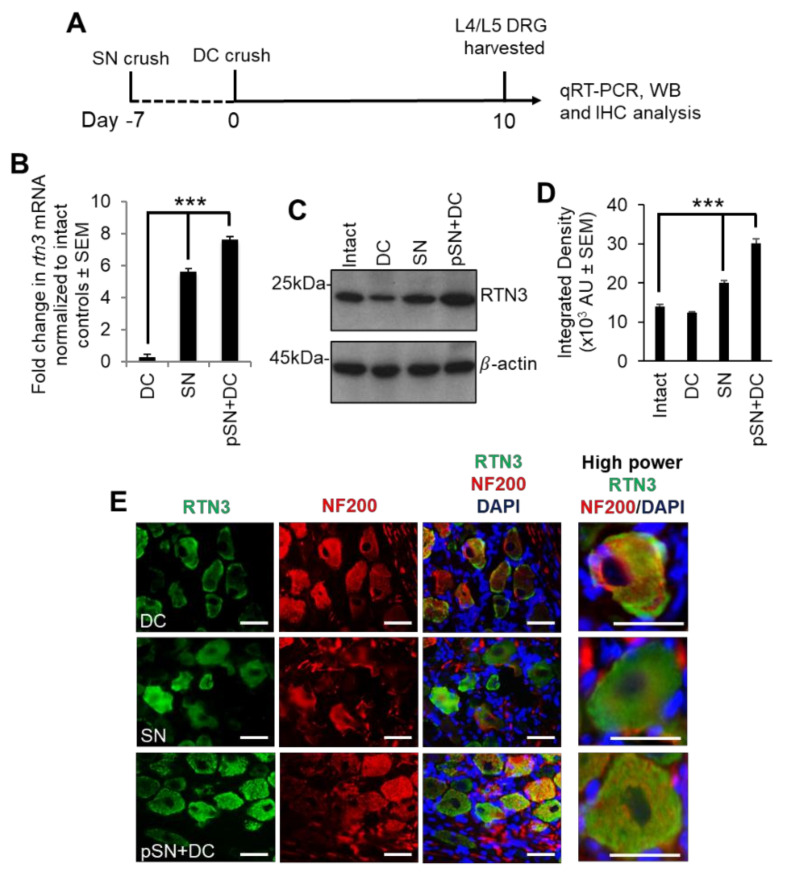
RTN3 is upregulated in dorsal root ganglion neurons (DRGN) after preconditioning lesions. (**A**) The diagram shows the timeline of treatments and tissue harvesting. (**B**) qRT-PCR to confirm upregulated RTN3 mRNA in SN and pSN + DC injury models (*n* = 16 DRG/group (8 rats/group)). (**C**) Western blot to show levels of RTN3 in control, DC, SN and pSN + DC injury models (*n* = 16 DRG/group (8 rats/group)). (**D**) Densitometry to confirm upregulation of RTN3 in SN and pSN + DC injury models (*n* = 16 DRG/group (8 rats/group)). (**E**) Immunohistochemistry to show immunolocalisation of RTN3 (green) in DRGN (red). Data are means ± SEM. Scale bars in (**E**) = 50 µm. *** *p* = 0.0001, one-way ANOVA with Dunnett’s post hoc test.

**Figure 2 cells-10-02015-f002:**
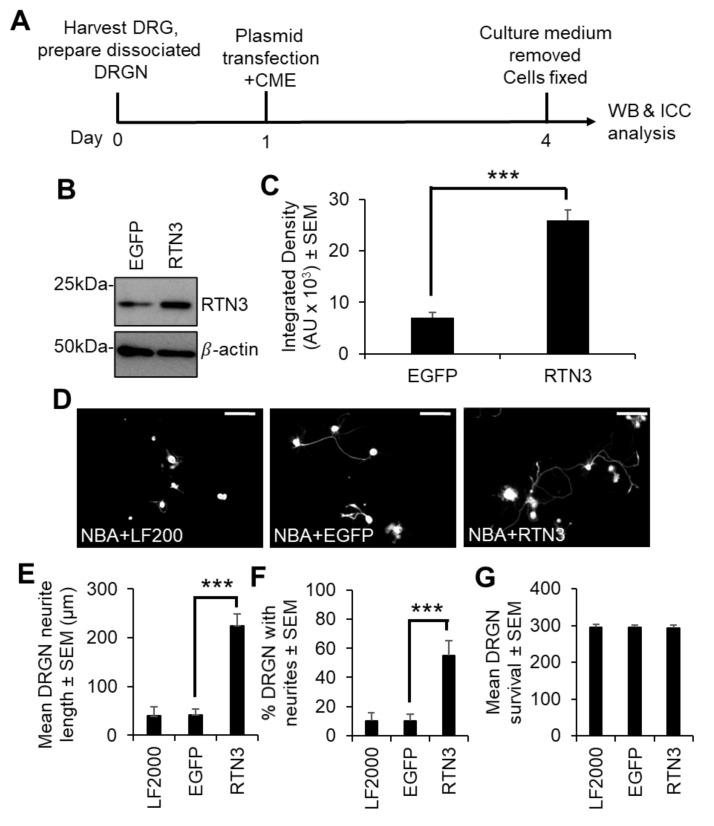
Overexpression of RTN3 in naïve DRGN cultures promotes neurite outgrowth. (**A**) Timeline of experiment and analysis methods. (**B**) Western blot to show RTN3 plasmids overexpress RTN3 protein. (**C**) Densitometry to show that RTN3 was significantly upregulated in RTN3 transfected cultures. (**D**) Representative images to show DRGN neurite outgrowth after RTN3 overexpression. RTN3 overexpression increases (**E**) the mean DRGN neurite length, (**F**) % DRGN with neurites but does not affect (**G**) DRGN survival. LF2000 = Lipofectamine 2000, AU = arbitrary units, *** *p* = 0.0001, one-way ANOVA with Dunnett’s post hoc test, *n* = 9 wells/treatment. Scale bars in (**D**) = 100 µm.

**Figure 3 cells-10-02015-f003:**
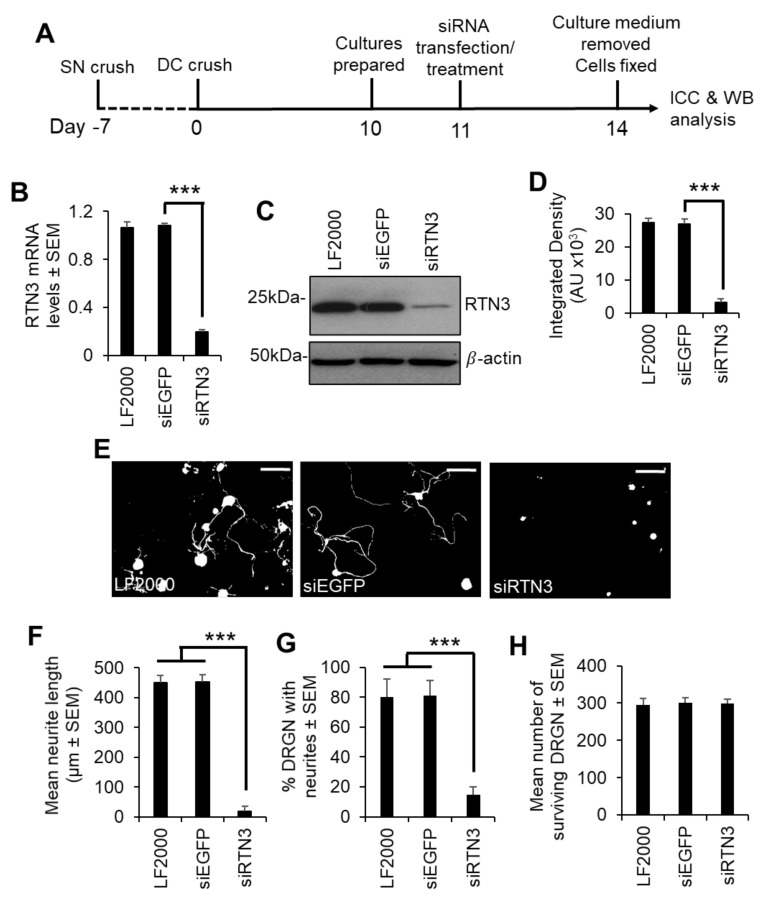
RTN3 is required for disinhibited DRGN neurite outgrowth. (**A**) Experimental timeline for analysis of RTN3 knockdown in preconditioned DRGN. (**B**) qRT-PCR to show ~80% knockdown of RTN3 using siRTN3. (**C**) Western blot to show levels of RTN3 protein after shRNA-mediated knockdown in DRGN. β-actin is used as a loading control. (**D**) Densitometry to show that PEI-shRTN3 significantly knocks down RTN3 protein in DRGN cultures. (**E**) Representative images to show neurite outgrowth after LF2000, siEGFP and siRTN3 treatment. (**F**) Quantification to show the mean longest neurite length after shRNA-mediated knockdown of RTN3. (**G**) Quantification to show % DRGN with neurites after shRNA-mediated knockdown of RTN3. (**H**) Quantification to show that DRGN survival is unaffected after shRNA-mediated knockdown of RTN3. Data are means ± SEM. Scale bars in (**E**) = 100 µm. *** *p* = 0.0001, one-way ANOVA with Dunnett’s post hoc test. *n* = 9 wells/condition.

**Figure 4 cells-10-02015-f004:**
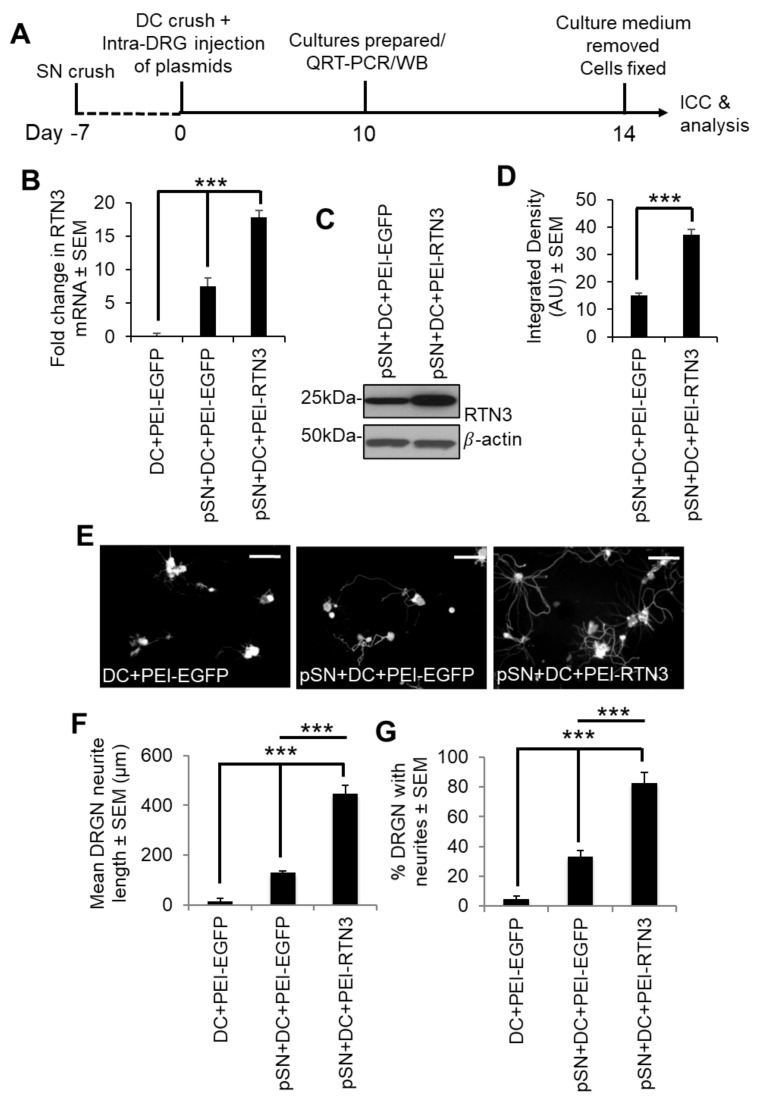
Overexpression of RTN3 promotes disinhibited DRGN neurite outgrowth. (**A**) Timeline for experimental analysis of RTN3 overexpression in preconditioned DRGN. (**B**) Fold-change in mRNA after RTN3 overexpression in preconditioned DRGN. (**C**) Western blot to show overexpressed levels of RTN3 protein. β-actin is used as a protein loading control. (**D**) Densitometry to show significant RTN3 overexpression by PEI-RTN3. (**E**) Representative images to show neurite outgrowth after DC + PEI-EGFP, pSN + DC + PEI-EGFP and pSN + DC + PEI-RTN3 treatment. (**F**) Quantification to show the mean neurite length after DC + PEI-EGFP, pSN + DC + PEI-EGFP and pSN + DC + PEI-RTN3 treatment (**G**) Quantification to show % DRGN with neurites after DC + PEI-EGFP, pSN + DC + PEI-EGFP and pSN + DC + PEI-RTN3 treatment. Data are means ± SEM. *** *p* = 0.0001, one-way ANOVA with Dunnett’s post hoc test. Scale bars in (**E**) = 100 µm. *n* = 9 wells/treatment.

**Figure 5 cells-10-02015-f005:**
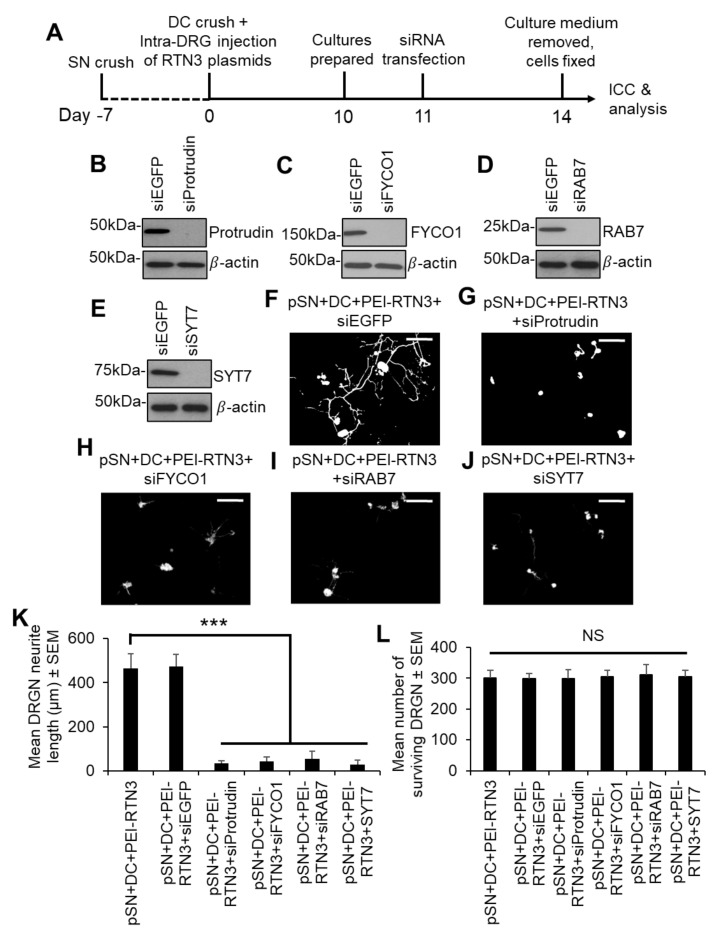
Knockdown of Protrudin and its interactors attenuates RTN3-induced DRGN neurite outgrowth. (**A**) Timeline for experimental analysis of Protrudin and its interactors in RTN3-overexpressed DRGN. Representative Western blots to confirm knockdown of (**B**) Protrudin, (**C**) FYCO1, (**D**) RAB7 and (**E**) SYT7. β-actin is used as a protein loading control. Representative images to show neurite outgrowth in (**F**) RTN3 overexpressed DRGN controls and after knockdown of (**G**) Protrudin, (**H**) FYCO1, (**I**) RAB7 and (**J**) SYT7 in RTN3-overexpressed cultures. (**K**) Quantification to show attenuation of RTN3-overexpressed DRGN neurite outgrowth by shProtrudin, shFYCO1, shRAB7 and shSYT7. (**L**) DRGN survival remains unaffected by knockdown of Protrudin, FYCO1, RAB7 and SYT7 in RTN3-overexpressed DRGN. *n* = 9 wells/condition Scale bars in (**F**–**J**) = 100 µm. Data are means ± SEM. *** *p* = 0.0001, one-way ANOVA with Dunnett’s post hoc test.

**Figure 6 cells-10-02015-f006:**
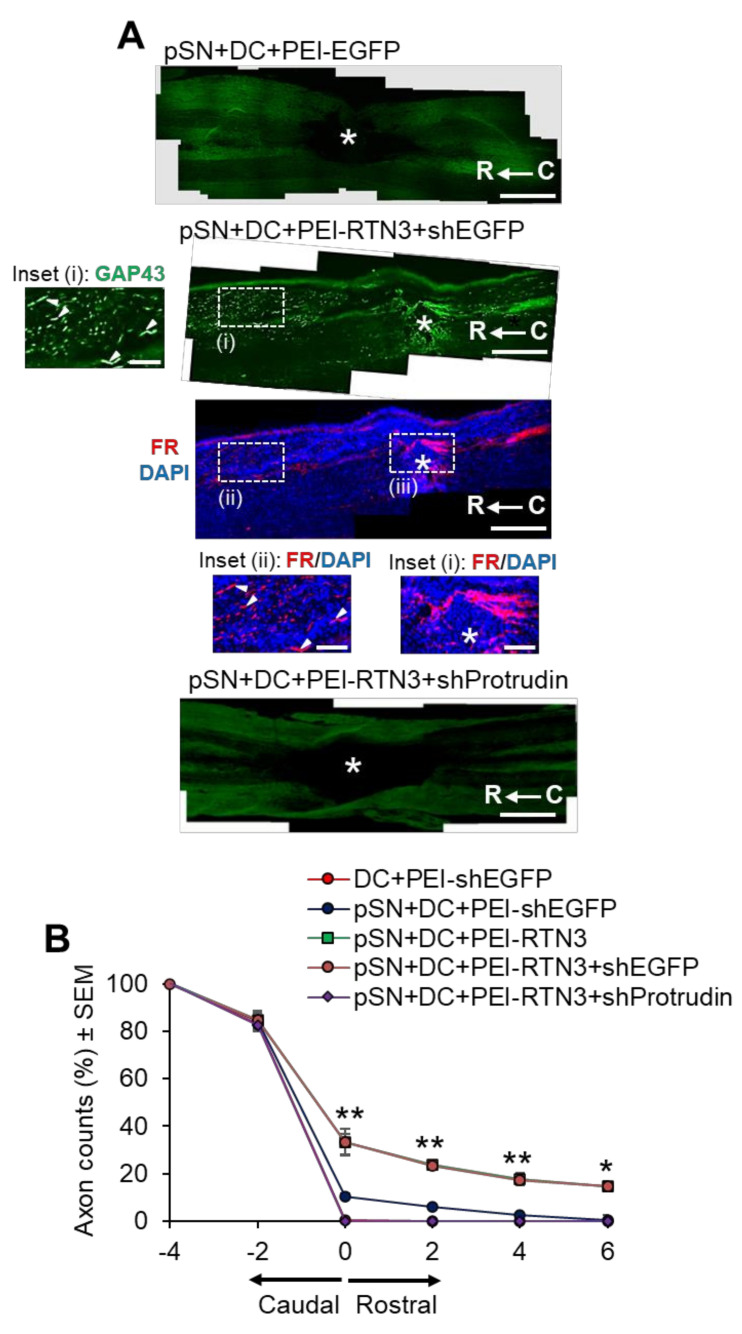
RTN3 overexpression promotes Protrudin-dependent DC axon regeneration. (**A**) Representative longitudinal sections of the spinal cord immunostained with GAP43 antibodies after treatment with pSN + DC + PEI-EGFP, pSN + DC + PEI-RTN3 + shEGFP and pSN + DC + PEI-RTN3 + shProtrudin (* = lesion site; C to R = caudal to rostral). FluoroRuby (FR) traced axons in the same sections as pSN + DC + PEI-RTN3 + shEGFP confirms overlap of GAP43^+^ axons (arrowheads), unequivocally demonstrating axon regeneration. Inset (i) shows high power images of GAP43^+^ axons in the boxed region (i). Inset (ii) and (iii) show high power images of FR^+^ axons in boxed regions (ii) and (iii). Note the overlap (arrowheads) between GAP43^+^ axons in inset (i) and FR+ axons in inset (ii). (**B**) Quantification of the % of GAP43^+^ axons at different distances caudal and rostral to the lesion site. Scale bars in (**A**) = 200 µm. *n* = 18 rats/group. * *p* = 0.05, ** *p* = 0.001, one-way ANOVA with Dunnett’s post hoc test.

**Figure 7 cells-10-02015-f007:**
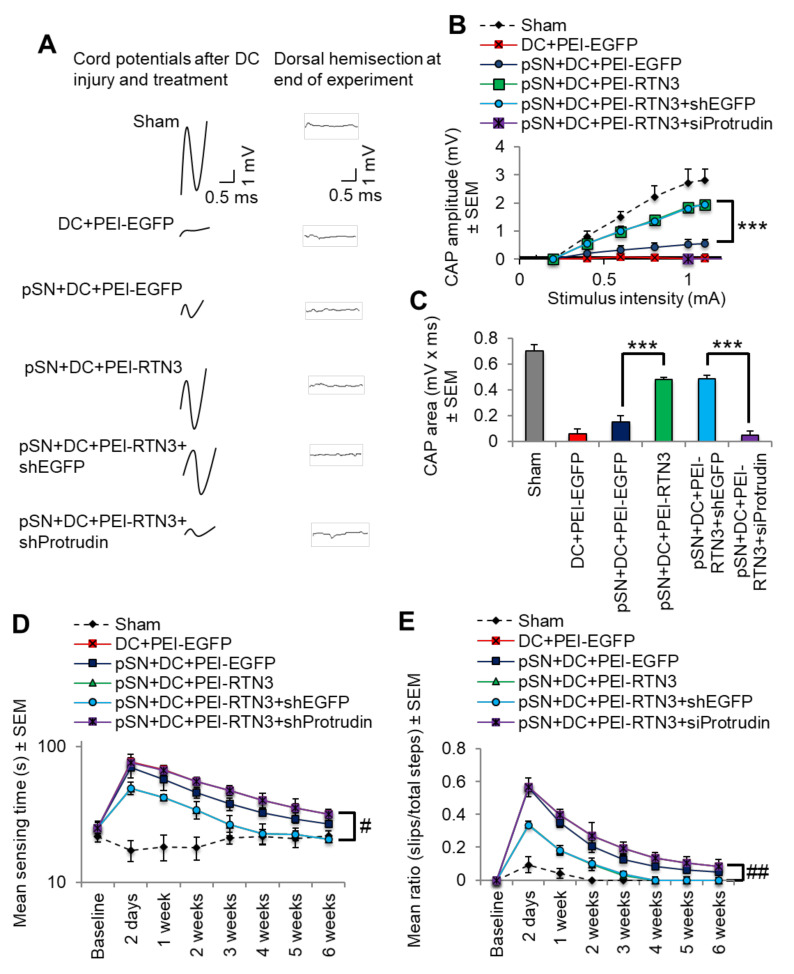
RTN3 overexpression promotes electrophysiological, sensory and locomotor functional recovery after DC injury. (**A**) Representative Spike 2 processed CAP traces in sham, DC + PEI-EGFP, pSN + DC + PEI-EGFP, pSN + DC + PEI-RTN3, pSN + DC + PEI-RTN3 + shEGFP and pSN + DC + PEI-RTN3 + shProtrudin-treated animals. (**B**) Negative CAP amplitudes at different stimulation intensities in sham, DC + PEI-EGFP, pSN + DC + PEI-EGFP, pSN + DC + PEI-RTN3, pSN + DC + PEI-RTN3 + shEGFP and pSN + DC + PEI-RTN3 + shProtrudin-treated animals. (**C**) CAP areas at all stimulation intensities in sham, DC + PEI-EGFP, pSN + DC + PEI-EGFP, pSN + DC + PEI-RTN3, pSN + DC + PEI-RTN3 + shEGFP and pSN + DC + PEI-RTN3 + shProtrudin-treated animals. (**D**) Mean tape sensing and removal times in sham, DC + PEI-EGFP, pSN + DC + PEI-EGFP, pSN + DC + PEI-RTN3, pSN + DC + PEI-RTN3 + shEGFP and pSN + DC + PEI-RTN3 + shProtrudin-treated animals. (**E**) Mean error ratio to show the number of slips over the total number of steps in sham, DC + PEI-EGFP, pSN + DC + PEI-EGFP, pSN + DC + PEI-RTN3, pSN + DC + PEI-RTN3 + shEGFP and pSN + DC + PEI-RTN3 + shProtrudin-treated animals. Data are means ± SEM. *n* = 18 rats/group. *** *p* = 0.0001, one-way ANOVA with Dunnett’s post hoc test. # *p* = 0.0012, linear mixed models (LMM); ## *p* = 0.00012, generalised linear mixed models (GLMM).

**Figure 8 cells-10-02015-f008:**
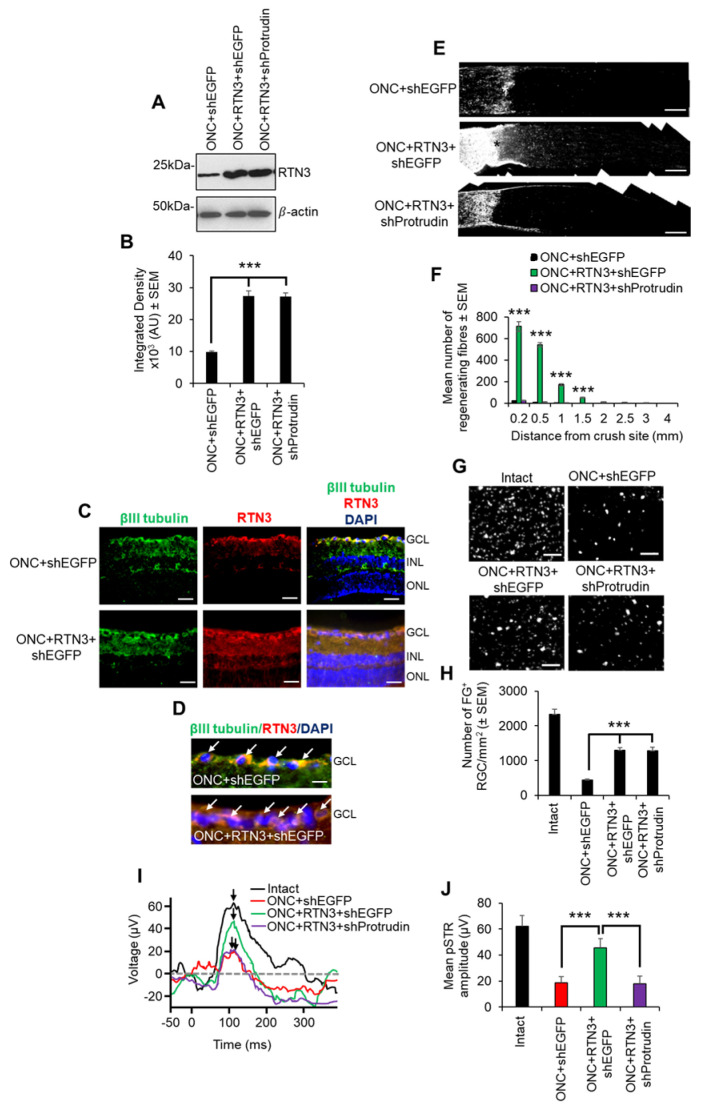
RTN3 overexpression promotes RGC axon regeneration and functional recovery after ONC, which is also protrudin-dependent. (**A**) Western blot and (**B**) quantification to show that RTN3 plasmids significantly upregulate RTN3 protein in the retina after intravitreal injection in vivo. (**C**) Immunohistochemistry to localise RTN3 protein to the RGCs (green) in the ganglion cell layer (GCL) and the inner plexiform and inner nuclear layer (INL). ONL = outer nuclear layer. (**D**) High power images to show RTN3 protein localised to βIII-tubulin^+^ RGCs (white arrows) in the GCL. (**E**) GAP43 immunohistochemistry (* = lesion site) and (**F**) quantification to show that in control optic nerve, few, if any, GAP43^+^ axons pass beyond the lesion site, whilst in RTN3 overexpressed eyes, significant GAP43^+^ axons are present beyond the lesion site, an effect which is obliterated when Protrudin is knocked out at the same time as RTN3 overexpression. (**G**) Representative images to show FluorGold (FG) backfilled RGC in retinal wholemounts after RTN3 overexpression. (**H**) Quantification of the number of FG^+^ RGC in retinal wholemounts shows that overexpression of RTN3 is significantly neuroprotective, an effect that is independent of protrudin. (**I**) Representative ERG traces and (**J**) quantification of the pSTR amplitude show significant improvements in RTN3 overexpressed eyes, which are ablated after protrudin knockdown. Black arrows show the peak of the pSTR. Data are means ± SEM. *n* = 12 eyes/optic nerves/treatment. Scale bars in (**C**) = 100 µm, scale bars in (**D**) = 25 µm, scale bars in (**E**) = 200 µm, scale bars in (**G**) = 50 µm. *** *p* = 0.0001, one-way ANOVA with Dunnett’s post hoc test.

**Figure 9 cells-10-02015-f009:**
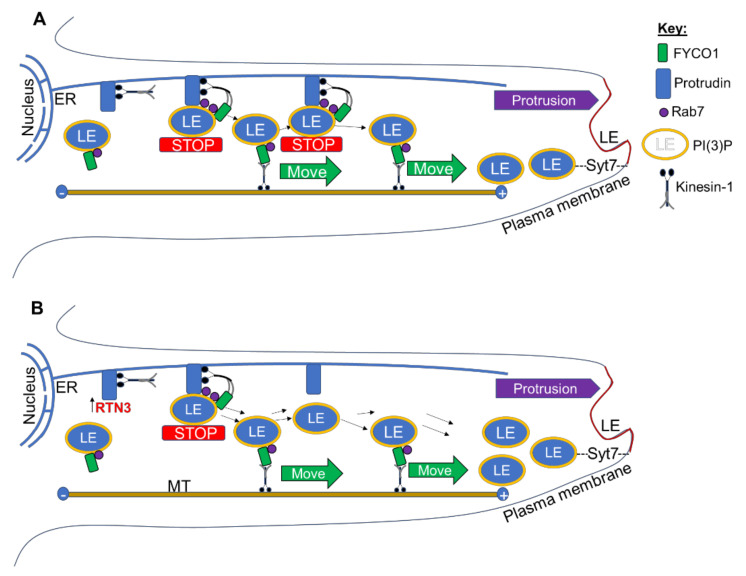
(**A**) Endoplasmic reticulum (ER)-localised protrudin forms contact sites with late endosomes (LEs) after detection of Rab7 and phosphoinositide 3 phosphate PI(3)P (yellow) and stopping the movement of LE (STOP). Kinesin-1 bound to protrudin is then handed over to the LE protein FYCO1, which mediates plus-end (+)-directed LE movement (MOVE) along microtubules (MT) to the periphery. Synaptotagmin 7 (SYT7)-mediated LE fusion with the plasma membrane then enables protrusion formation and neurite outgrowth, as previously proposed [14,57]. (**B**) Overexpression of RTN3 allows the handover of kinesin-1 to occur more rapidly and hence greater numbers of LEs available for protrusion (neurites/axons) at the plasma membrane.

## Data Availability

Source data for all figures in this manuscript, including unpropped blots are provided in Source Data File 1.

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
