# Peer review of "Overexpression of Reticulon 3 Enhances CNS Axon Regeneration and Functional Recovery after Traumatic Injury"

_cells, 2021, doi:10.3390/cells10082015_

Round 1

Reviewer 1 Report

Dr. Alhajlah S et al. revealed that inhibition of RTN3 resulted in the promotion of axonal regeneration and neuronal survival after injuries. This paper seems to be valuable used methods, and results and their estimations are reasonable.  

Author Response

Comment: Dr. Alhajlah S et al. revealed that inhibition of RTN3 resulted in the promotion of axonal regeneration and neuronal survival after injuries. This paper seems to be valuable used methods, and results and their estimations are reasonable.  

Author response: Thank you. No changes required.

Reviewer 2 Report

Title should be "after traumatic injury" to exclude ischemic or inflammatory injury.

Author analyzed neuronal protection and regrowth. How about the effect for glia cells and immunological status?   

Author Response

Comment: Title should be "after traumatic injury" to exclude ischemic or inflammatory injury.

Author response: The title has been amended accordingly.

Comment: Author analysed neuronal protection and regrowth. How about the effect for glia cells and immunological status?

Author response: Here we have not analysed the response of glial cells and immunological status after RTN3 overexpression, however, these are important considerations that we are currently analysing. Nonetheless, the effects on glia and immunological status will not affect the eventual results on DC axon regeneration that we observed.     

Reviewer 3 Report

In this interesting manuscript the authors evaluate how the overexpression of reticulon 3 could enhance CNS axon regeneration and functional recovery after injury.

As the authors stated, various intrinsic and extrinsic factors (e.g. Reticulon 4 or Nogo-A) could act as inhibitory factors for axons regeneration.

The authors could briefly speculate in the discussion section, as they mention in the conclusions, how their findings can be evaluated in relation to the previous researches on this topic in order to delineate a sort of unified theory of CNS axon regeneration and summarize future lines of research and potential therapeutic approaches.

Author Response

Comment: The authors could briefly speculate in the discussion section, as they mention in the conclusions, how their findings can be evaluated in relation to the previous researches on this topic in order to delineate a sort of unified theory of CNS axon regeneration and summarize future lines of research and potential therapeutic approaches.

Author response: We have provided a new figure for our proposed working model and have related discussion to previous work.

Reviewer 4 Report

We read with great interest the article by Alhajlah et al where they assessed the role of

 Reticulon 3-mediated axonal regeneration and studied the mechanism involved including the role of protrudin which was successfully evaluated in both spinal cord and optic nerve injury. The work has implications for regenerative medicine and rehabilitation that can be implemented in traumatic brain injury.

The work is well designed where the authors have applied different approaches to demonstrate the contribution of Reticulon 3 and protrudin cross talk.

The work involves high throughput data (utilizing microarray); the data should be presented, and the gene changes should be presented as well. The data are of limited use and should be exploited to show other involved genes in the spinal cord and optic nerve injury. It is mentioned that the Data were analyzed using GeneSpring GX7; please show these data.  

For the functional recovery, the authors should elaborate on the selected behavioral testing performed in terms of how rapid the functional recovery is was obtained discussing the temporal dynamics of Reticulon 3 and protrudin contribution to this kind recovery.

Final suggested work:

It would be very important to evaluate inflammation in the glial scarring in terms of GFAP expression due to increased axonal regeneration; if not feasible, it would be great to  have it in the discussion

Minor comments:

The authors should review the discussion section as they have a non-relevant section:

Authors should discuss the results 821 and how they can be interpreted from the perspective of previous studies and of the working hypotheses. The findings and their implications should be discussed in the broadest context possible. Future research directions may also be highlighted

 Please correct these sentences:

Sample sizes were no computed using power calculations, instead, these are based on previous similar experiments published by us.

Also, correct some places with double parentheses.

Author Response

Comment 1: The work involves high throughput data (utilizing microarray); the data should be presented, and the gene changes should be presented as well. The data are of limited use and should be exploited to show other involved genes in the spinal cord and optic nerve injury. It is mentioned that the Data were analysed using GeneSpring GX7; please show these data. 

Author response: The microarray were performed by a collaborator and postdoc in the laboratory in 2007. However, we only had access to a finalised list of 12 genes that were highly regulated in regenerating and non-regenerating scenarios. We are therefore unable to provide the original GeneSpring data as the collaborator PI has since retired and is not contactable. We have therefore removed all reference to microarray studies in our manuscript. Despite the lack of this microarray the data showing changes in RTN3 are still valid as they have been performed independently of the microarray and a verified by qPCR, western blot and IHC. We are extremely sorry that we cannot provide this data. We will continue to contact the collaborator and the postdoc to try and obtain the original GeneSpring files. 

Comment 2: For the functional recovery, the authors should elaborate on the selected behavioral testing performed in terms of how rapid the functional recovery is was obtained discussing the temporal dynamics of Reticulon 3 and protrudin contribution to this kind recovery.

Author response: We have added a couple of lines in the discussion to cover these points.

Comment 3: It would be very important to evaluate inflammation in the glial scarring in terms of GFAP expression due to increased axonal regeneration; if not feasible, it would be great to have it in the discussion.

Author response: We agree that this is important. However, it is not feasible as Dr Alhajlah who performed these experiments as part of his PhD has moved on to a different laboratory several years ago. However, we are planning to follow the original study with looking at inflammation, scarring and apoptosis after RTN3 overexpression. We have now added a few sentences in the discussion to mention these points.

Comment 4: The authors should review the discussion section as they have a non-relevant section:

Author response: We have reviewed the discussion and made sure that everything is now relevant.

Comment 5: Authors should discuss the results and how they can be interpreted from the perspective of previous studies and of the working hypotheses. The findings and their implications should be discussed in the broadest context possible. Future research directions may also be highlighted.

Author response: We have now added a figure of our proposed model and have discussed this with mention of previous findings and have made this as broad as possible.

Comment 6: Sample sizes were no computed using power calculations, instead, these are based on previous similar experiments published by us.

Author response: Corrected

Comment 7: Also, correct some places with double parentheses.

Author response: We have checked our use of double parentheses and they appear correct to us.

Round 2

Reviewer 4 Report

Accept